# A Field Investigation on Adaptive Thermal Comfort in an Urban Environment Considering Individuals' Psychological and Physiological Behaviors in a Cold-Winter of Wuhan

**Mehdi Makvandi [1]**, **Xilin Zhou [1],[2],* **, **Chuancheng Li [1],* ** and **Qinli Deng [1]**

[1] School of Civil Engineering and Architecture, Wuhan University of Technology, Wuhan 430070, China; Mehdi_makvandi@whut.edu.cn (M.M.); deng4213@whut.edu.cn (Q.D.)

[2] Department of Architecture and Building Engineering, School of Environment and Society, Tokyo Institute of Technology, Yokohama 2268502, Japan

* Correspondence: zhou.xilin@whut.edu.cn or zhou.x.ag@m.titech.ac.jp (X.Z.); licc@whut.edu.cn (C.L.); Tel.: +86-18986174097 (X.Z.); +86-13477032166 (C.L.)

**Abstract:** To date, studies of outdoor thermal comfort (OTC) have focused primarily on physical factors, tending to overlook the relevance of individual adaptation to microclimate parameters through psychological and physiological behaviors. These adaptations can significantly affect the use of urban and outdoor spaces. The study presented here investigated these issues, with a view to aiding sustainable urban development. Measurements of OTC were taken at a university campus and in urban spaces. Simultaneously, a large-scale survey of thermal adaptability was conducted. Two groups were selected for investigation in a cold-winter-and-hot-summer (CWHS) region; respondents came from humid subtropical (Cfa) and hot desert (BWh) climates, according to the Köppen Climate Classification (KCC). Results showed that: (1) neutral physiological equivalent temperature (NPET) and preferred PET for people from the Cfa (PCfa) and BWh (PBWh) groups could be obtained with KCC; (2) PCfa adaptability behaviors were, subjectively, more adjustable than PBWh; (3) Clothing affected neutral temperature (NT), where NT reduced by approximately 0.5 °C when clothing insulation rose 0.1 Clo; and (4) Gender barely affected thermal acceptance vote (TAV) or thermal comfort vote (TCV) and there was a substantial relationship between thermal sensation, NT, and PET. These findings suggest 'feels like' temperature and comfort may be adjusted via relationships between microclimate parameters.

**Keywords:** outdoor thermal comfort (OTC); effective environmental elements; behavioral conformity; Cfa and BWh; cold-winter-and-hot-summer

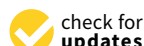



## 1. Introduction

In recent years, the increased attention paid to energy-saving initiatives and public well-being has led to a greater focus on understanding thermal comfort, which is an essential factor in promoting the use of urban public spaces [1,2]. In the standard set by the American Society of Heating, Refrigerating, and Air-Conditioning Engineers (ASHRAE), thermal comfort is defined as a level of human satisfaction obtained through thermal surroundings [3]. Depending on the subject's location, thermal comfort can be classified as outdoor thermal comfort (OTC), semi-OTC thermal comfort, and indoor thermal comfort [4]. Some thermal studies [5,6] have concentrated on indoor thermal comfort where thermal conditions are comparatively fixed, and others [7,8] have focused on locations with unstable surrounding conditions and on OTC in urban open spaces.

In research focused on OTC, many studies have focused on physical factors [9–11]. Such studies have been carried out in urban canyons [12], pedestrian areas [13], parks [14], residential communities [15], and across various climates such as hot desert [16], hot-humid [17], humid subtropical [18], cold semi-arid [19], cold [14], and severely cold

climates [20]. A range of thermal comfort indices have been used by researchers including the outdoor standard effective temperature (OUT_SET*) [17], predicted mean vote (PMV) [21], predicted percentage dissatisfied (PPD) [22], physiological equivalent temperature (PET) [23], universal thermal climate index (UTCI) [24], wet bulb globe temperature (WBGT) [25], and the thermal sensation vote (TSV), as indicated within the seven-point scale in ASHRAE [26]. The PET index is one of the most popular thermal indices designed for the investigation of thermal parameters outdoors. It integrates the impacts of all physiologically-related climate parameters (relative humidity, air temperature, mean radiant temperature, and wind speed) and personal parameters (metabolic rate and clothing insulation level) to give a single temperature index [27].

These thermal studies specify the extent to which microclimate and its effects influence human thermal comfort levels in open spaces. However, very few scholars have focused on the adaptability of individuals [28] to the microclimate parameters [29] that affect OTC, and there has been little consideration of psychological and physiological behaviors in light of the Köppen Climate Classification (KCC) [30] and geographical divisions [31]. These factors can significantly affect urban areas that are used for outdoor activities and interactions. Therefore, the adaptability of people, as a phenomenon influenced by geographic location and urban climate and in light of the KCC, demands greater attention in OTC studies.

The consideration by city-makers of outdoor activities [32], and OTC [33] plays a key role in evaluating urban climate [34] and can significantly affect the physical, psychological, and social health of individuals in urban spaces. In this context, more voices are now calling for the inclusion of outdoor activities that will generate better health outcomes as an element in sustainable urban development strategies [35].

On a university campus, students and others frequently work indoors for extended periods. This means that outdoor activities become highly significant for them, in terms of ensuring their sustained good health [36]. Such campuses often contain a diverse range of people and given that diverse people, with different thermal histories, may have different neutral temperatures (NT) [37] and acceptable temperature ranges (ATR) [27], there is clearly a need for more research concerning the physical and psychological behaviors that may affect OTC and for these to be investigated in light of the KCC system.

On campuses, thermal studies of human comfort have been conducted both indoors and outdoors [38,39]. For instance, Ghaffarianhoseini et al. [40] studied outdoor thermal (dis)comfort in different zones on a campus during the noontime period in a tropical region. They found that the mean of partially shaded and completely shaded spaces generated PET values at the upper end of the thermal comfort zone, (range 26–30 °C) and increased thermal comfort range (22–34 °C—considering the adaptation). They also evaluated the conditions required to increase user comfort out of doors, and noted that shaded spaces could promote thermal comfort. Indeed, their ability to do this exceeded that of unshaded spaces by about 30%.

Scholars have also examined thermal comfort within university office spaces [37]. Li and Liu [41] reviewed hundreds of outdoor thermal studies and identified Out-SET*, UTCI, PET, and PMV as the four leading indices for thermal comfort; these scholars also investigated the interaction of thermal comfort and human performance, and the impact of the former upon the latter. In this review study, the 'neutral PET/NPET' range was 24–29 °C, and this was supported by more than 90% of the case studies. In thermal comfort studies, Shahzad [37] suggested using the term 'preferred temperature' in preference to NT, where the NT is usually represented as a comfortable temperature range.

Studies have also been conducted on thermal comfort and environmental parameters in single office buildings on a university campus [42,43]. Liang et al. [44] compared UTCI and PET indices alongside numerical outcomes for outdoor thermal surroundings in exterior campus spaces. They found an inherent difference between PET and UTCI, and indicated that these could not be used interchangeably. They also stated that the PET index gives an enhanced range of sensation temperatures, and can predict thermal comfort for the outdoor environment including important thermal benchmarks such as NT and ATR. In a

similar outdoor thermal study, Cheung and Jim [27] evaluated ATR (the range approved by more than 80% of respondents) as ranging from 22.6 to 25.4 °C, PET ranging from 17.0 to 31.9 °C, and UTCI ranging from 19.0 to 33.0 °C.

Some thermal studies have evaluated psychological and physiological responses [45, 46], and several thermal analyses have illuminated the significance of adaptive thermal comfort and the performance [47,48]. However, as far as the authors of this study are aware, previous studies have been rarely focused on people's psychological and physiological behaviors in light of the KCC system directly. In particular, there is little evidence concerning the effects of the simultaneous interaction among the various factors that affect OTC. Furthermore, few studies have made a simultaneous comparison between the university campus setting and other major, frequently-used spaces in urban environments. Such investigation would be helpful in understanding public thermal comfort and its effect on the users of such spaces.

With the objective of closing this gap in the literature, the authors chose the Wuhan University of Technology (WHUT) campus, with its surrounding urban spaces, as the location for the study presented here. The campus is located in Wuhan, China. Wuhan is geographically located in a CWHS region. The study site had not previously been evaluated for thermal comfort using the KCC system, nor had it been used to study OTC focusing on people from both humid subtropical (PCfa) and hot desert (PBWh) climates. The study presented here aims to understand the thermal comfort of individuals from various population groups in the outdoor space on Wuhan's campus, using the KCC system to assess the physical and psychological adaptive behaviors that affect OTC.

The primary advantage of using the KCC method in an OTC study is the KCC's enhanced ability to obtain accurate thermal comfort ranges such as NPET and the preferred PET when focusing on multiple and diverse groups. This study used comfort temperature (CT) with related urban physical and meteorological parameters, ATR, PBWh, and PCfa, to study individuals' physical and psychological conformity behaviors. It takes into account gender, clothing (in relation to NT), and the significant relationships between microclimate parameters, in order to draw conclusions about OTC. Additional studies are required to thoroughly understand the natures and interaction of influential factors. These will help to improve OTC design strategies related to urban planning and design.

Thus, the main objectives of the study presented here are:

(1)  To investigate outdoor thermal performance parameters and their characteristics in various locations in a CWHS zone to identify key factors that influence OTC.
(2)  To investigate individual understanding of thermal matters in exterior (outdoor) parts of the university campus at Wuhan, analyzing the outcomes in light of prior thermal comfort studies in outdoor spaces.
(3)  To examine the impacts of coming from a different climate (PCfa and PBWh) and of biological gender as well as clothing insulation on thermal comfort in open spaces, and to examine dissimilar human performance in relation to environmental conformity to obtain OTC.

For this study, the authors used combined statistical techniques [49–53] to link averaged thermal sensation vote, thermal comfort vote, thermal acceptability vote, thermal acceptability rate, and physiological equivalent temperature with neutral temperature zone, comfort temperature range, and acceptable temperature scope for two groups of people with different biometeorological backgrounds. The study methodology also included a questionnaire survey [54], thermal images [55], and standard parametric devices [23] for on-site meteorological measurements. Meteorological statistics were also used.

## 2. Materials and Methods

### 2.1. Study Area

The west campus of Wuhan University of Technology (30°31′19.3″N, 114°20′55.2″E), along with its surrounding urban spaces, are located in the Hongshan area. This lies in the central part of Wuhan City, in China's Hubei Province.

Building design in China can be classified according to five main climate divisions, based on the mean temperatures of the hottest and coldest months, which are July and January, respectively (Figure 1) [56]. Hence, the Wuhan metropolis area is in a CWHS region, which is notable for this study's objectives. In this area, the average temperature of the hottest month is between 27.9 and 34.6 °C, and the average temperature of the coldest month is between 3 and 10.5 °C; the sum of days with a daily average temperature lower than 7 °C is between 45 and 98, and the sum of days with the daily average temperature higher than 28 °C is between 60 and 120.

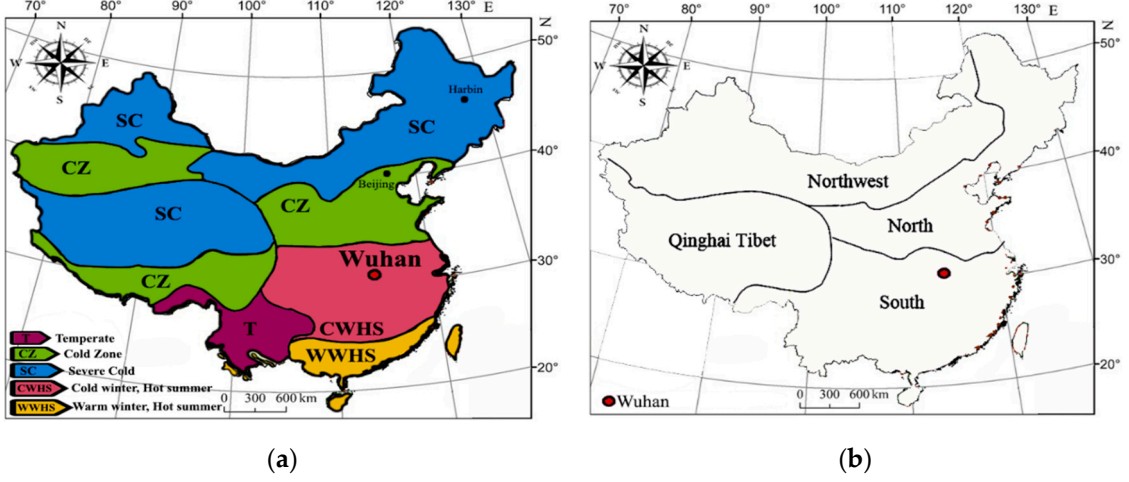

(**a**)  (**b**)

| Climate Regions | Monthly Temperature in Average | |
|---|---|---|
| | Lowest Air Temp. | Highest Air Temp. |
| SC | −10 °C ≥ | 25 °C ≥ |
| CZ | −10–0 °C | 18–28 °C |
| T | 0–13 °C | 18–25 °C |
| CWHS | 0–10 °C | 25–30 °C |
| WWHS | 10 °C < | 25–29 °C |

(**c**)

**Figure 1.** Geographic and climate zones in China. (**a**) Climate regions in China; (**b**) geographical divisions in China; (**c**) identified climate zone average monthly temperature under the classification range.

In terms of local weather data (Figure 2), the average outdoor air temperature ranges from 3.1 °C to 34.6 °C, with the maximum monthly mean temperatures in July (39 °C) and August (36 °C), and the minimum monthly mean temperatures in January (−2 °C) and February (1 °C). Relative humidity (RH) varies slightly across the year; the range is between 71.2% and 78.2%, with a mean annual RH of 75%.

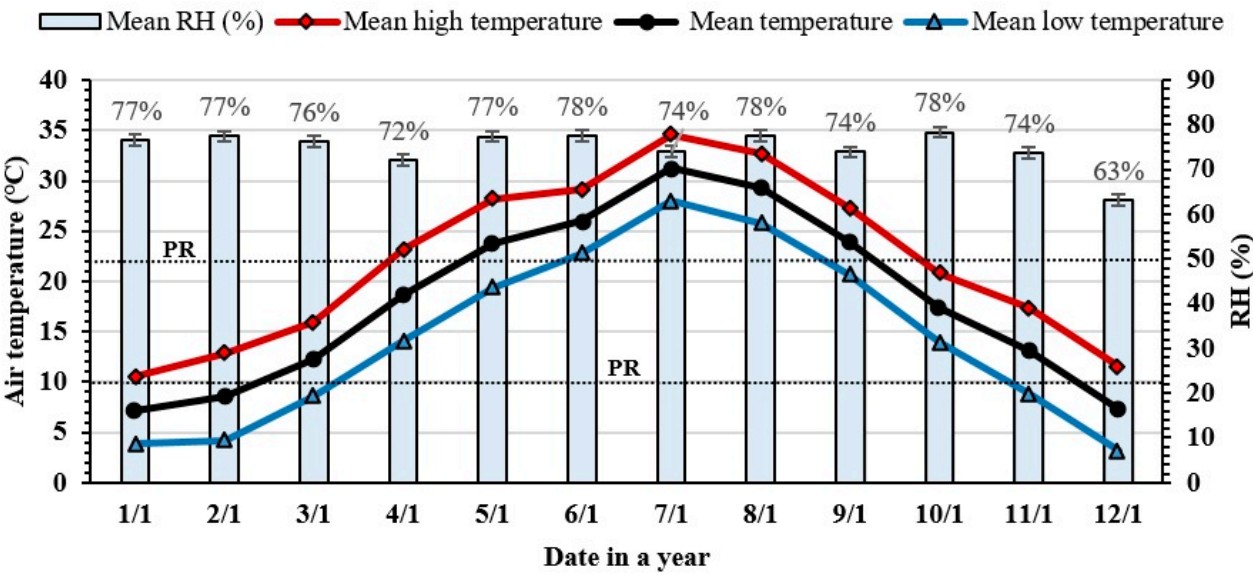

**Figure 2.** The monthly variation of air temperature (Ta) and relative humidity (RH) in Wuhan (2010–2018).

Figure 3 shows the fluctuations in mean daily temperature, mean low temperature, and the mean high temperature for an ordinary year in Wuhan. Based on the seasonal climatic division by the pentad (5-day mean) [57] mean temperature method—according to the China Meteorological Administration (CMA) [58]—the first day of a pentad that has a mean temperature below 10 °C constitutes the start of winter, the range 10–22 °C is designated the period of autumn or spring. The first day of a pentad that has a mean temperature of more than 22 °C throughout is considered the start of summer.

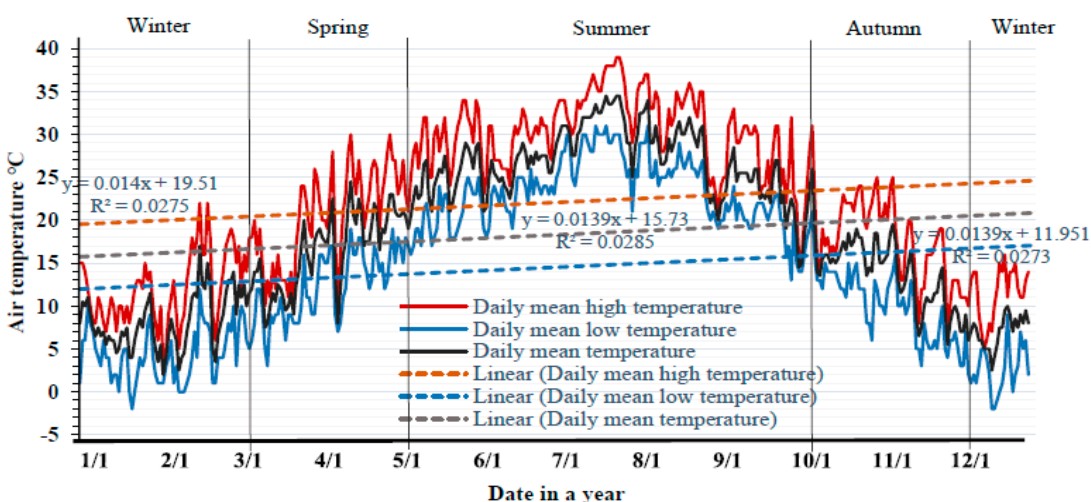

**Figure 3.** The daily temperature fluctuations in an ordinary year in Wuhan.

Using the pentad method and its range (PR), it is clear that the summer in Wuhan lasts five months, from the initial days of May to the latest days of September, and the periods of both autumn (from October to November) and spring (March to April) are no longer than four months combined. The winter period lasts three months and runs from early December to the end of the following February. Winter and summer are thus notable and extended periods, suitable for the investigation of human thermal comfort and the adaptive behaviors undertaken by people at the university campus of WHUT, many of whom come from outside Wuhan.

Figure 4 shows the locations used for measurements in this study. The west campus of WHUT is divided into four main parts. The two first parts are located inside the campus, and the third part is on the main road, with traffic effects, lying just between the west and east campuses in which the intensity of urban heat island (UHI) [12] is higher. The fourth part is next to the campus in a residential area with a high-rise building effect [15], which is a frequently-used space with a substantial impact on the urban microclimate. A combination of methods was used for this investigation. The first technique was a fixed-point observation [18] used to measure on-site reference data in an open space and at the selected locations. The second technique was mobile observation [59] used to collect data essential for a more tangible comparison between frequently-used spaces in urban areas (specifically, the residential area; high-rise and mid-rise buildings, road; typical traffic level, and open space area) and the campus study area. In the latter area, most consideration points are characterized as having urban parameters that influence thermal comfort. The third technique comprised an on-site questionnaire survey [54] for those of both genders. Respondents came from two groups, namely those from Cfa (Wuhan's local people) and BWh (African; hot desert) climates. The fourth technique, used to collect temperature data for surfaces and participants, was to use a digital infrared thermometer gun and hand-held infrared (IR) thermal camera [60]. These methods were used to collect crucial thermal comfort data as benchmarks to improve the evaluation of environmental and thermal open space acceptability.

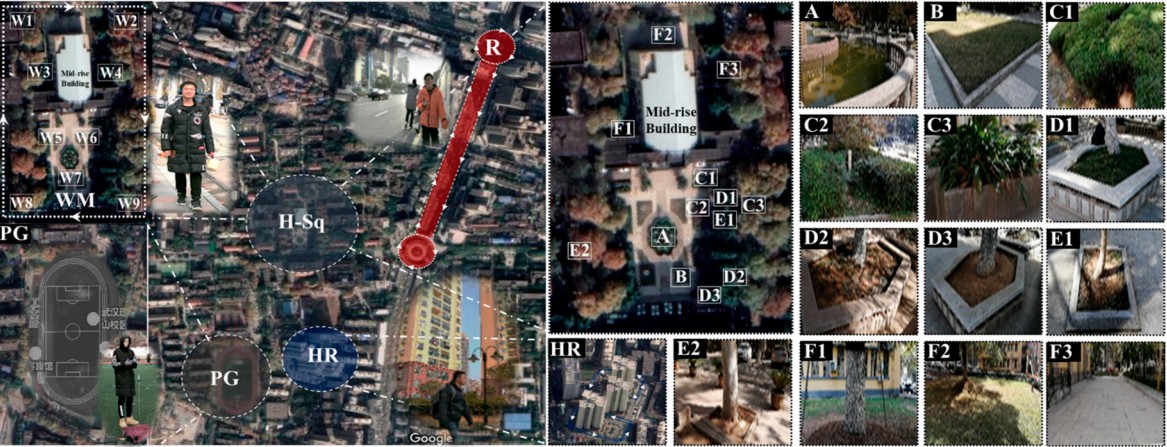

**Figure 4.** Measurement points and locations for fixed and mobile observation.

## 2.2. Data Collection

In order to avoid climate conditions that would be inappropriate for carrying out surveys and measurements, the data collection was completed on sunny days in winter and continued for seven consecutive days to achieve precision. The daily average air temperatures of the designated winter days fell in the range of winter monthly average air temperatures, therefore the chosen days accurately signify cold days during a normal year. Measurements were taken between 8:00 a.m. and 8:00 p.m., with data collection carried out within three significant times of day, based on the impact of UHI on the specified parameters. These times of day were morning (8:00–10:00 a.m.), afternoon (1:00–3:00 p.m.) and evening (6:00–8:00 p.m.). In all, twenty-five sites were chosen for on-site measurement, along with influential surfaces nearby. In this research, all factors influencing thermal surroundings were documented including relative humidity (RH), air temperature (Ta), globe temperature (GT), solar radiation (SR), and wind velocity (Va) [23].

Table 1 shows the characteristics of the main digital instruments used in thermal environmental measurement. All devices complied with ISO (International Organization for Standardization) 7726 [61]. During the investigation, all device data were recorded

automatically in an identified time range, whereby the recording times for RH and Ta were every 10 s, SR was every 10 min, and Va and GT were every 1 min at a height of 1.5 m. Table 2 explains the ambiance of the surroundings for each measurement point. Although investigations on thermal comfort usually concentrate either on peak winter or peak summer, the present investigation was not carried in peak summer due to the outbreak of Covid-19 in Wuhan but rather in peak winter. Additionally, several studies have focused on the middle season, therefore, even if the study was not done in the summer peak, meaningful outcomes could still be achieved [20,62,63].

**Table 1.** The characteristics of the main instruments used in thermal environmental measurement.

| Parameter Measured | Sensor | Range | Accuracy | Name of Instruments |
|---|---|---|---|---|
| Relative humidity (RH) | Polymer membrane | 15% to 95% | $\pm2.5\%$ | HOBO Temp/RH logger (U-series datalogger) |
| Air temperature (Ta) Dew point (Dwpt) | thermistor | $-20\ °C$ to $70\ °C$ | $\pm0.21\ °C$ | HOBO Temp/RH logger (U-series datalogger) |
| Body temperature (BT) | Thermal Sensor | $-20\ °C$ to $300\ °C$ | $\pm2\ °C$ | FLIR C3 camera |
| Solar radiation (SR) | Silicon Photodiode | 0 to $2000\ W/m^2$ | $\leq\pm2\%$ | JTR05 |
| Globe temperature (GT) | Metallic Globe | $-10°$ to $+100\ °C$ | $\pm0.1\ °C$ | LY-09 |
| Wind velocity (Va) | Hot-wire Anemometer | 0 to $40\ m/s$ | $\pm0.2\ °C$ $0.1\ m/s$ | TES-1341 |

**Table 2.** Measurement points with surroundings ambiance.

| No | Marked Points | Measurement Point Descriptions | Surroundings Ambiance |
|---|---|---|---|
| 1 | A | Water Area/Water Body | Lawn |
| 2 | B | Low-Rise Grass Coverage | Bushes and water |
| 3 | $C_1$ | High-Dense Bushes | Pavement |
| 4 | $C_2$ | Mid-Dense Bushes | Small trees |
| 5 | $C_3$ | Low-Dense Bushes | Pavement |
| 6 | $D_1$ | High-Rise Tree ($HRT_1$) | High Dense Grass |
| 7 | $D_2$ | High-Rise Tree ($HRT_2$) | Spread grass |
| 8 | $D_3$ | High-Rise Tree ($HRT_3$) | Soil |
| 9 | $E_1$ | Low-Rise Tree | Soil |
| 10 | $E_2$ | High-Rise Tree ($HRT_4$) | Soil and pedestrian |
| 11 | $F_1$ | Mid-Rise Tree ($MRT_1$) | Grass, soil, bushes close to the building |
| 12 | $F_2$ | Pavement | The building, small trees, mid-dense bushes |
| 13 | $F_3$ | Mid-Rise Tree ($MRT_2$) | Grass, soil, bushes away of building |
| 14 | $W_1$ | Northwest Corner of The Site | Pedestrian |
| 15 | $W_2$ | Northeast Corner of The Site | Pedestrian |
| 16 | $W_3$ | Green Spaces $_1$ (Tree, Lawn, Bushes) | Building |
| 17 | $W_4$ | Green Spaces $_2$ (Tree, Lawn) | Building and pavement |
| 18 | $W_5$ | Lawn $_1$ | Building |
| 19 | $W_6$ | Lawn $_2$ | Building and bushes |
| 20 | $W_7$ | Pavement | Water body |
| 21 | $W_8$ | Southwest Corner of The Site | Pedestrian under shadow |
| 22 | $W_9$ | Southeast Corner of The Site | Pedestrian |
| 23 | R | Road | Urban Structures/commercial buildings |
| 24 | HR | High-Rise Buildings | Concrete walls |
| 25 | PG | Playground | Roadway |

### 2.3. Questionnaire Survey

A questionnaire study and analysis of human thermal comfort and the behavioral conformity of PCfa and PBWh groups was conducted alongside assessment of physical

and psychological climate impacts. The questionnaire survey, shown in Table 3, comprised six main sections. The first section requested fundamental information such as age, weight, height, and sex. The second section, based on the Köppen Climate Classification (KCC) [30], asked about the climate in regions where the participants had lived in the past five years [64]. The third section sought information on the current climate conditions in which respondents lived and explored subjective and physical adaptation. The fourth section explored participant activity and clothing including recent activity and metabolic rate (W/m$^2$), and considered the impact of clothing insulation level (Clo) [23] on the human body. During the survey, each participant's body temperature was measured and the thermal camera applied. All measurements complied with standard 55 (thermal environmental circumstances) [65] and 7730 [66] specified by ASHRAE and ISO. The fifth section of the questionnaire comprised thermal analysis concerning TSV, TCV, and TAV [64], which are, respectively, the thermal sensations vote, thermal comfort vote, and thermal acceptability vote. The sixth section examined alterations and adaptive behaviors influenced by the prevailing environment where comfort was the goal sought.

The examined contributors were mostly PCfa and PBWh students and local inhabitants, who had been in the outdoor environment for at least 10–20 min [64] at the marked measurement points. Five hundred and thirty-six questionnaires were completed; there were 413 PCfa respondents and 123 PBWh respondents. The respondents' ages were mostly between 18 and 24 (74% of PCfa) and 31 to 40 (41% of PBWh) years old. For PCfa respondents, male respondents were 52% and female respondents were 48%, while male and female PBWh respondents were 59% and 41%, respectively. Statistical tests were used to estimate the required sample value to calculate effect sample values for the questionnaire analysis. The estimate revealed that more than 110 responses were needed for each type, and over 220 in all to ensure the study's accuracy [67–69]. Thus, according to the estimation, the sample amounts were sufficient to perform an accurate and useful study.

**Table 3.** The questionnaire used in this study.

| Section | Category | Subcategory | | | | | | |
|---|---|---|---|---|---|---|---|---|
| Section 1 | Basic Info. | Biological Gender | ☐ Male | | | ☐ Female | | |
| | | Age | ☐ < 18 | ☐ 18–24 | ☐ 25–30 | ☐ 31–40 | ☐ 41–50 | ☐ 51–60 | ☐ >60 |
| | | Weight (kg) | ☐ < 50 | ☐ 50–60 | ☐ 60–70 | ☐ 70–80 | ☐ 80–90 | ☐ 90–99 | ☐ >99 |
| | | Height (cm) | ☐ <155 | ☐ 155–160 | ☐ 160–170 | ☐ 170–180 | ☐ 180–190 | ☐ >190 |
| Section 2 | Climate regions where participants lived before | Köppen Climate Classification (KCC)  | ☐ Af | ☐ Am | ☐ Aw | ☐ BWh | ☐ BWk | ☐ BSh |
| | | | ☐ BSk | ☐ Csa | ☐ Csb | ☐ Csc | ☐ Cwa | ☐ Cwb |
| | | | ☐ Cwc | ☐ Cfa | ☐ Cfb | ☐ Cfc | ☐ Dsa | ☐ Dsb |
| | | | ☐ Dsc | ☐ Dsd | ☐ Dwa | ☐ Dwb | ☐ Dwc | ☐ Dwd |
| | | | ☐ Dfa | ☐ Dfb | ☐ Dfc | ☐ Dfd | ☐ ET | ☐ EF |
| Section 3 | New Climate Conditions for non-Chinese | Adaptation | ☐ Yes | | | ☐ No | | |
| | | How Long/ Year | ☐ <1 | ☐ 1–2 | | ☐ 2–3 | ☐ >3 | |
| | | Dissimilarities | ☐ Very Similar | ☐ Similar | ☐ Neutral | ☐ Different | | ☐ Very Different |
| | | Physical and Psychological Changes | ☐ Yes | | | ☐ No | | |
| | | | If "Yes" what is the effect. | | | ☐ Behavior problem | | |
| | | | ☐ Health problems | | | ☐ Health and behavior problem | | |
| Section 4 | Individual Activity and Clothing | Recent Activity and Metabolic Rate (W/m2) | ☐ Seated relaxed (58) | | | ☐ Standing, light activity (93) | | |
| | | | ☐ Walking (110) | | ☐ Exercise (360) | | | ☐ Running (500) |
| | | | Upper Part: | | ☐ T-shirt (0.15) | | | ☐ Long-sleeved Shirt (0.19) |
| | | | ☐ Short-sleeved Shirt (0.25) | | | ☐ Thermal Underwear (0.1) | | |

**Table 3.** *Cont.*

| | | | | | |
|---|---|---|---|---|---|
| Section 4 | Individual Activity and Clothing | Clothing Insulation Level (Clo) on Human Body | ☐ Knitwear (0.28) | ☐ Hoodie (0.3) | ☐ Jacket (0.35) |
| | | | ☐ Woolen Jacket (0.45) | ☐ Wadded Jacket (0.5) | ☐ Down Jacket (0.55) |
| | | | Lower Part: ☐ Shorts (0.08) | | ☐ Thermal Underwear (0.1) |
| | | | ☐ Thin Skirt (0.15) | ☐ Dress (0.2) | ☐ Thick Skirt (0.25) |
| | | | ☐ Thin Trouser (0.24) | ☐ Thick Trouser (0.28) | |
| | | | Feet: ☐ Thin Socks (0.02) | ☐ Thick Socks (0.05) ☐ Boots (0.08) | |
| | | | ☐ Leather Shoes (0.06) | ☐ Slippers (0.02) | ☐ Sandal (0.02) |
| Section 5 | Thermal Analysis | Thermal Sensations Vote (TSV) | ☐ −3 (Very Cold) | ☐ −2 ( Cold) | ☐ −1(Cool) |
| | | | ☐ −0.5( Slightly Cool) | ☐ 0 (Neutral) | ☐ 0.5 (Slightly Warm) |
| | | | ☐ 1 (Warm) | ☐ 2 (Hot) | ☐ 3 (Very Hot) |
| | | Thermal Comfort Vote (TCV) | ☐ −2 (Very Uncomfortable)  ☐ −1 (Slightly Uncomfortable) | ☐ 0 (Comfortable)  ☐ 1 (Slightly Comfortable) | ☐ 2 (Very Comfortable) |
| | | Thermal Acceptability Vote (TAV) | ☐ Quite Unacceptable (1) | ☐ Just Unacceptable (2) | |
| | | | ☐ Quite Acceptable (3) | ☐ Just Acceptable (4) | |
| Section 6 | Alterations | Adaptive Behaviors (AB) | ☐ Wearing Gloves | ☐ Utilizing Umbrella | ☐ Wearing a Hat |
| | | | ☐ Staying Under the Sun | ☐ Going to the shaded place | |
| | | | ☐ Taking off clothes | ☐ Putting on Extra Clothes | ☐ No Changes |

### 2.4. Computation of the Indices for Outdoor Thermal Comfort (OTC)

2.4.1. Physiological Equivalent Temperature Index (PETI)

Physiological equivalent temperature is a universal climate index that is used to investigate thermal comfort, which assesses the thermal component of the surroundings/outdoor microclimate. It is based on the Munich Energy-balance Model for Individuals (MEMI) [70]. It is a two-node model that models the human body's thermal surroundings in a physiologically relevant way, and is assessed under the complex outdoor conditions (applicable for both significant studies related to steady-state and unsteady-state environments). Physiological equivalent temperature is described as the Ta; the heat budget of the individual body is balanced with the same core and skin temperature as under the complex outdoor circumstances to be evaluated [71].

Applying the index above-mentioned has advantages for any outdoor study that has a thermo-physiological basis. It provides the actual impact of the environment's sensation on humans, and is of practical use in hot and colder climates. It depends on the significant climatology factors irrespective of clothing (Clo values) and metabolic activity (Met values) [72]. Determination of the mean radiant temperature (MRT) [23] is required to estimate the physiological equivalent temperature, which is a significant meteorological variable governing the human heat balance [73] and thermal comfort in outdoor spaces. MRT is defined as the exchange of short and longwave radiation between a human and its surrounding environment; its equation is given below [30]:

$$MRT = \left[ (GT + 273.15)^4 + \frac{(1.10 \times 10^8 V_a^{0.6})(GT - T_a)}{\varepsilon D^{0.4}} \right]^{0.25} - 273.15 \tag{1}$$

where *MRT* is the mean radiant temperature, °C; $T_a$ is the air temperature, °C; *GT* is globe temperature, °C; $V_a$ is the air velocity, m/s; *D* is the standard globe diameter (*D* = 0.15 m), m; and $\varepsilon$ is the absorption value of the globe ($\varepsilon$ = 0.95).

In this research, the physiological equivalent temperature value was determined through climatological parameters (Ta, RH, SR, Va, and MRT) using the RayMan model [74]. This model is typically applied to evaluate outdoor environments and has been verified by several investigations [75,76].

2.4.2. Dew Point Calculation Using Relative Humidity (RH)

Given the connection of dew point (DewPt, °C) with thermal comfort, its temperature can be calculated from RH as formulated below [77]:

$$T_{DewPt} = \frac{B_1 \cdot \left[ \ln\left( \frac{RH}{100} \right) + \frac{A_1 \cdot T_a}{B_1 + T_a} \right]}{A_1 - \ln\left( \frac{RH}{100} \right) - A_1 \cdot \frac{T_a}{B_1 + T_a}} \tag{2}$$

where $T_{DewPt}$ is the dew point temperature; *Ta* is the ambient temperature; *RH* is the percent relative humidity; $A_1$ is a constant equal to 17.625; and $B_1$ is a constant equal to 243.04 °C. The following equation gives the formula for measuring RH based on the study pattern [78]:

$$RH = \frac{W}{W_S} \times 100\% \tag{3}$$

where the *RH* equation calculates the relative humidity based on the actual density of vapor (*W*) and the saturated density ($W_S$).

### 2.5. Evaluating Data Using Statistical Methods

In the research presented here, various statistical approaches were implemented to evaluate and analyze the data including logistic regression [50], multiple linear regression [51], and T-test [52]. Linear regression was employed to find linkages between averaged thermal sensation vote (ATSV), averaged thermal comfort vote (ATCV), averaged

thermal acceptability vote (ATAV), thermal acceptability rate (TAR), and physiological equivalent temperature index (PETI) to specify neutral temperature zone (NTZ), comfort temperature range (CTR), and acceptable temperature scope (ATS), respectively. T-test was applied to define whether there were notable correlations in thermal sensation vote (TSV), thermal comfort vote (TCV), and thermal acceptability vote (TAV) between genders, groups (PCfa and PBWh), and influential surrounding parameters at specified points in the study. Logistic regression was applied to discover the possibility of different physical and psychological behavior adaptations with the alteration of physiological equivalent temperature.

## 3. Results and Analysis

### 3.1. Outdoor Thermal Surroundings

Thermal data regarding maximum, minimum, mean, and standard deviation (SD) are given in Tables S1–S3 in part A of the Supplementary Materials. The mean globe temperature and solar radiation of the environment are given in Figure 5. The analysis revealed that the mean values of Ta, GT, and SR at points C1, D2, D3, C2, F1, and F3 were much lower than those of other points because these measurement points were under the crowns of trees and bushes. The mean RH of points C1, C2, F1, and F3 was greater and more significant than those of other points. The lowest temperature change was found in bare soil and water—as effective environmental parameters—with a Ta range of 0.3 °C and 0.4 °C, respectively. A comparison of effects of the significant landscape elements of water, bare soil, and soil covered by grass, is shown in Figure 6.

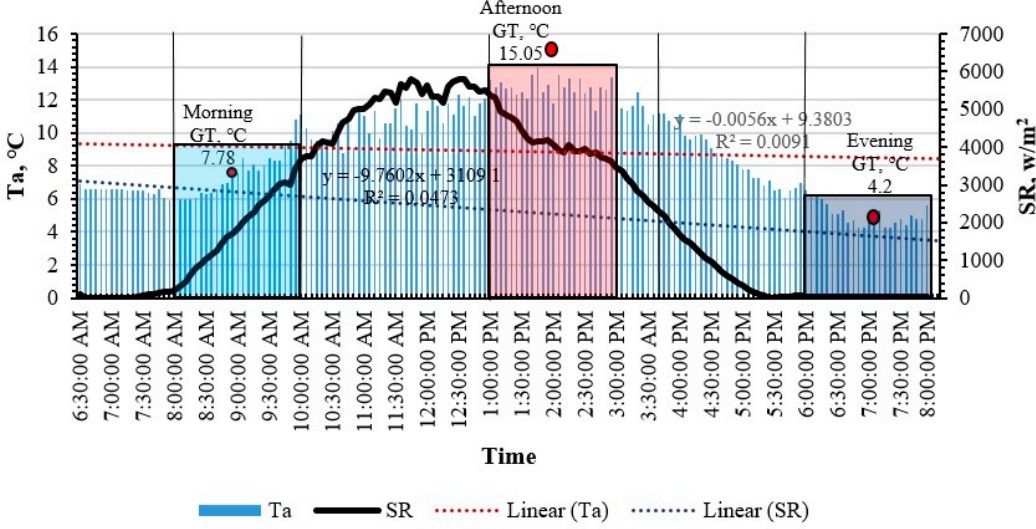

**Figure 5.** The mean solar radiation (SR), environmental temperature (Ta), and globe temperature (GT) in the measured area.

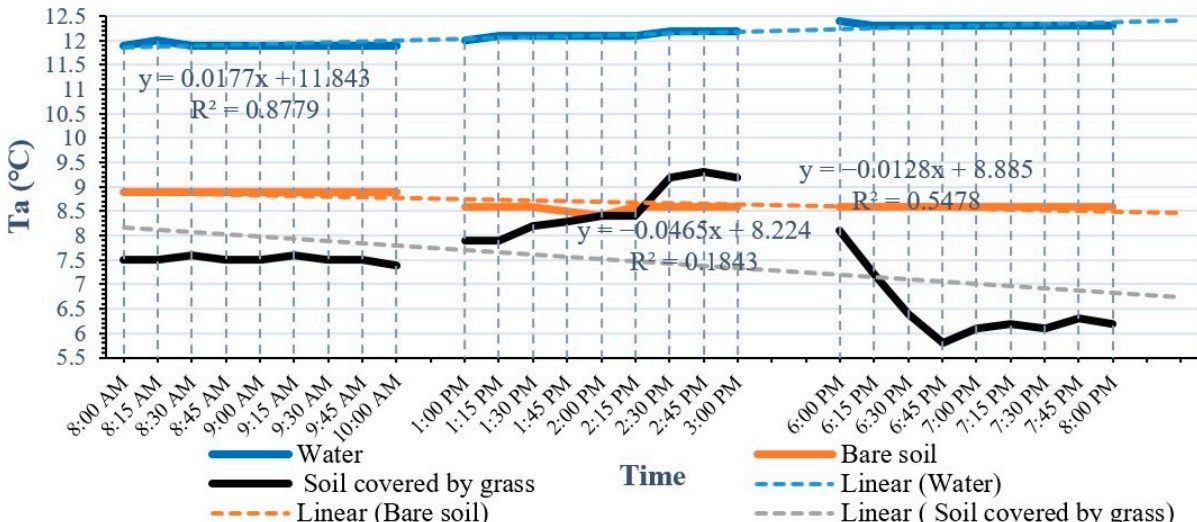

**Figure 6.** A comparison effect between significant landscape elements of water, bare soil, and soil covered by grass.

A strong relationship between the Ta and RH parameters of all measured points within three main time zones (morning, afternoon, and evening) are shown in part B of the Supplementary Materials (Figure S1). This significantly identified a connection between physical arrangement and landscape patterns, which can adjust OTC both physically and psychologically. Accordingly, the proportion of space occupied by water, and its combination with landscape arrangements, can generate more stable-state conditions in terms of OTC, the effect being stronger with dense vegetation than with fragmented green spaces.

The results also show that the measured points' impervious surfaces, especially roads with asphalt material, pavements, mid-rise buildings, and high-rise buildings with concrete materials, augmented the surface temperature of the surroundings, which may increase MRT and perceptions of thermal comfort in cold conditions. However, it is essential to combine these elements with dense greenery and features containing water to regulate RH and Ta parameters and provide better OTC for all seasons. To understand the interactive relationship and effects of the leading outdoor thermal parameters, Ta and RH factors were compared with the DewPt parameter at the measurement points at various times; the findings are shown in Figure 7. It is clear that DewPt has a special relationship with the 'feels like' temperature and the sensation of human comfort, which also has a significant correlation with RH. When DewPt moves down, the RH goes down and vice versa.

Tables 4 and 5 reveal the mean radiant temperature and physiological equivalent temperature value of all measurement points in the main designated time zones; for obtained outdoor MRT values, corresponding 'PETI' was calculated via the Berkeley CBE Thermal Comfort Tool [79]. The overall average MRT of all points in the morning, afternoon, and evening were 10.06 °C, 26.64 °C, and 1.77 °C, respectively; the overall mean PETI of all points in the morning, afternoon, and evening were 4.4 °C, 11.8 °C, and 4.18 °C, respectively.

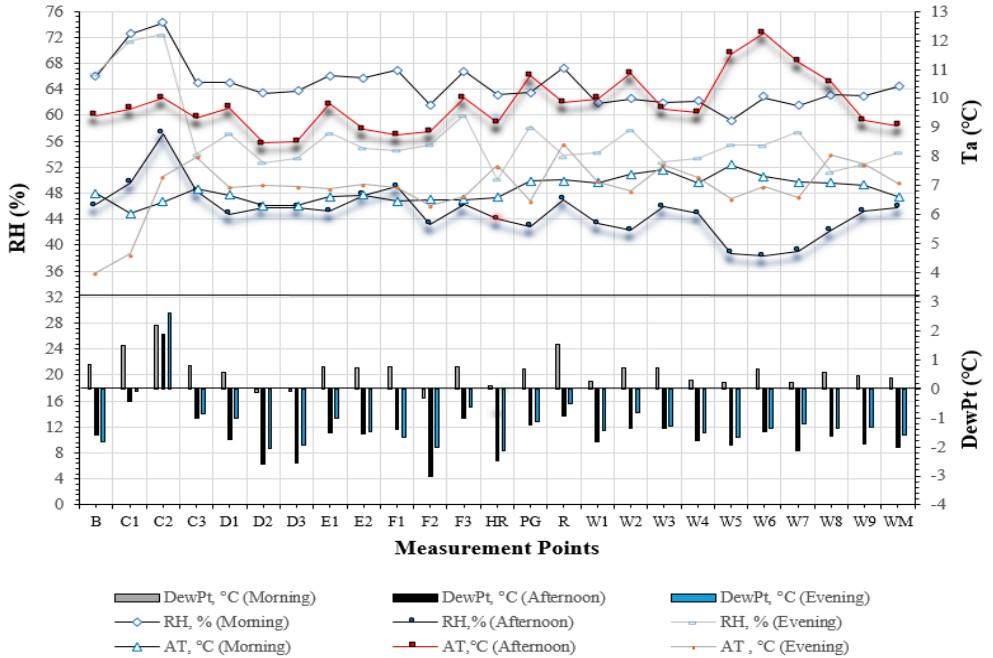

**Figure 7.** The interaction/relationship between the significant outdoor thermal parameters (Ta; air temperature, RH; relative humidity, and DewPt; dew point) in the measurement points within various time zones.

**Table 4.** Calculated mean radiant temperature (MRT).

| Measuring Points | | MRT/Mean Radiant Temperature | | | | | | | | | | | |
|---|---|---|---|---|---|---|---|---|---|---|---|---|---|
| | | Morning | | | | Afternoon | | | | Evening | | | |
| | | Mean | Max | Min | SD | Mean | Max | Min | SD | Mean | Max | Min | SD |
| 1 | B | 10.3 | 45.6 | 4.9 | 9.82 | 27.5 | 43.7 | 14.8 | 6.44 | 4.4 | 3.8 | 1.6 | 0.66 |
| 2 | C1 | 12 | 49 | 5 | 10.51 | 27 | 48.4 | 14.7 | 7.58 | 3.9 | 3.5 | 5.1 | 0.37 |
| 3 | C2 | 11 | 47.7 | 4.9 | 10.29 | 26.2 | 46.5 | 14.6 | 7.18 | 1.4 | 3 | −6 | 2.13 |
| 4 | C3 | 9.9 | 45.8 | 4.8 | 9.93 | 27.6 | 49.7 | 14.7 | 7.87 | 0.8 | 3 | −9.4 | 2.94 |
| 5 | D1 | 10.5 | 48 | 4.9 | 10.42 | 26.9 | 46.5 | 14.7 | 7.13 | 1.8 | 3.1 | −4.2 | 1.73 |
| 6 | D2 | 11.4 | 50 | 4.9 | 10.84 | 29.4 | 53.3 | 14.9 | 8.62 | 1.7 | 3.1 | −2.9 | 1.40 |
| 7 | D3 | 11.3 | 50 | 4.9 | 10.85 | 29.3 | 53.3 | 14.9 | 8.62 | 1.7 | 3.1 | −2.2 | 1.22 |
| 8 | E1 | 10.6 | 47.6 | 4.9 | 10.30 | 26.6 | 45.1 | 14.7 | 6.81 | 1.8 | 3.1 | −2.2 | 1.23 |
| 9 | E2 | 10.5 | 47.3 | 4.9 | 10.24 | 28.4 | 51.7 | 14.8 | 8.29 | 1.7 | 3.1 | −4.2 | 1.72 |
| 10 | F1 | 11 | 50.1 | 4.9 | 10.90 | 28.9 | 52.6 | 14.8 | 8.49 | 1.8 | 3.1 | −1.6 | 1.08 |
| 11 | F2 | 10.9 | 49.6 | 4.9 | 10.78 | 28.6 | 50.8 | 14.8 | 8.07 | 2.3 | 3 | −1 | 0.95 |
| 12 | F3 | 10.9 | 49.3 | 4.9 | 10.71 | 26.2 | 44.6 | 14.6 | 6.72 | 2.1 | 3.1 | −3.1 | 1.48 |
| 13 | HR | 10.7 | 48.5 | 4.9 | 10.52 | 28 | 47 | 14.9 | 7.17 | 1.1 | 3 | −4.1 | 1.63 |
| 14 | PG | 9.3 | 28.2 | 4.9 | 5.50 | 24.5 | 35.1 | 14.6 | 4.56 | 2.2 | 3.2 | −1.2 | 1.03 |
| 15 | R | 9.2 | 31.8 | 4.8 | 6.44 | 26.6 | 45.6 | 14.7 | 6.93 | 0.4 | 2.9 | −17.5 | 4.95 |
| 16 | W1 | 9.5 | 42.7 | 4.8 | 9.18 | 26.3 | 39.7 | 14.7 | 5.56 | 1.6 | 3.1 | −3.3 | 1.49 |
| 17 | W2 | 8.7 | 38.1 | 4.8 | 8.09 | 24.5 | 35.1 | 14.6 | 4.56 | 1.9 | 3.1 | −1.2 | 0.99 |
| 18 | W3 | 8.4 | 29.5 | 4.8 | 5.93 | 27 | 48 | 14.7 | 7.48 | 1.1 | 2.9 | −3 | 1.34 |
| 19 | W4 | 9.5 | 21.4 | 4.8 | 3.80 | 27.3 | 47 | 14.8 | 7.21 | 1.4 | 3 | −1.9 | 1.11 |
| 20 | W5 | 8 | 33.3 | 4.8 | 6.94 | 23 | 31.8 | 14.5 | 3.84 | 2.1 | 3.2 | 0.4 | 0.63 |
| 21 | W6 | 8.9 | 38.1 | 4.8 | 8.07 | 21.5 | 26.2 | 14.5 | 2.62 | 1.7 | 3.1 | 0.95 | 0.49 |
| 22 | W7 | 9.4 | 41 | 4.9 | 8.74 | 23.7 | 34.5 | 14.6 | 4.43 | 2.1 | 3.1 | −0.6 | 0.85 |
| 23 | W8 | 9.5 | 44.4 | 4.8 | 9.62 | 25.2 | 35.6 | 14.7 | 4.64 | 0.7 | 2.9 | −6.5 | 2.19 |
| 24 | W9 | 9.7 | 43.3 | 4.8 | 9.31 | 27.8 | 48.9 | 14.8 | 7.65 | 1 | 3 | −5 | 1.85 |
| 25 | WM | 10.6 | 47.3 | 4.9 | 10.23 | 28.2 | 50.6 | 14.8 | 8.04 | 1.6 | 3.1 | −8.2 | 2.73 |
| Average | | 10.06 | 42.7 | 4.86 | 9.12 | 26.64 | 44.45 | 14.71 | 6.66 | 1.77 | 3.1 | −3.25 | −3.25 |

**Table 5.** Calculated physiological equivalent temperature (PET).

| Measuring Points | | PETI/Physiological Equivalent Temperature | | | | | | | | | | | |
|---|---|---|---|---|---|---|---|---|---|---|---|---|---|
| | | Morning | | | | Afternoon | | | | Evening | | | |
| | | Mean | Max | Min | SD | Mean | Max | Min | SD | Mean | Max | Min | SD |
| 1 | B | 4.4 | 7.2 | 7 | 0.69 | 11.9 | 13.4 | 11 | 0.54 | 3.8 | 4.8 | 1.4 | 0.78 |
| 2 | C1 | 4.5 | 7.4 | 6.9 | 0.69 | 11.9 | 13.7 | 11.1 | 0.59 | 3.9 | 5.3 | 1.4 | 0.88 |
| 3 | C2 | 4.4 | 7.3 | 7 | 0.71 | 11.8 | 14 | 11.1 | 0.65 | 3.9 | 6.2 | 1.3 | 1.09 |
| 4 | C3 | 4.4 | 7.2 | 7.1 | 0.71 | 11.9 | 13.7 | 11.1 | 0.59 | 4.4 | 6.1 | 1.1 | 1.13 |
| 5 | D1 | 4.5 | 7.3 | 7 | 0.68 | 11.9 | 13.6 | 11 | 0.59 | 4.2 | 6 | 1.2 | 1.08 |
| 6 | D2 | 4.4 | 7.4 | 6.9 | 0.71 | 11.9 | 13.3 | 11.2 | 0.48 | 4.2 | 6 | 1.2 | 1.08 |
| 7 | D3 | 4.4 | 7.4 | 7 | 0.72 | 11.9 | 13.3 | 11.2 | 0.48 | 4.2 | 6 | 1.2 | 1.08 |
| 8 | E1 | 4.4 | 7.3 | 7 | 0.71 | 11.9 | 13.5 | 11 | 0.56 | 4.2 | 6 | 1.2 | 1.08 |
| 9 | E2 | 4.5 | 7.3 | 7 | 0.68 | 11.8 | 13.5 | 11.2 | 0.53 | 4.3 | 6 | 1.2 | 1.08 |
| 10 | F1 | 4.5 | 7.3 | 7 | 0.68 | 11.9 | 13.4 | 11.3 | 0.48 | 4.2 | 6 | 1.2 | 1.08 |
| 11 | F2 | 4.4 | 7.3 | 7 | 0.71 | 11.9 | 13.3 | 11.2 | 0.48 | 4.1 | 5.6 | 1.2 | 0.99 |
| 12 | F3 | 4.4 | 7.3 | 7 | 0.71 | 11.9 | 13.7 | 11 | 0.61 | 4.2 | 5.9 | 1.2 | 1.06 |
| 13 | HR | 4.4 | 7.3 | 7 | 0.71 | 11.8 | 13.3 | 11 | 0.52 | 4.3 | 6.2 | 1.2 | 1.12 |
| 14 | PG | 4.5 | 6.8 | 7 | 0.62 | 11.8 | 13.7 | 10.7 | 0.67 | 4.1 | 5.8 | 1.3 | 1.01 |
| 15 | R | 4.4 | 6.8 | 7.1 | 0.66 | 11.9 | 13.6 | 11 | 0.59 | 4.4 | 6.1 | 1.1 | 1.13 |
| 16 | W1 | 5 | 7.1 | 7 | 0.53 | 11.9 | 13.5 | 10.8 | 0.60 | 4.2 | 6 | 1.2 | 1.08 |
| 17 | W2 | 4.4 | 7 | 7 | 0.67 | 11.8 | 14 | 10.7 | 0.72 | 4.2 | 6 | 1.3 | 1.05 |
| 18 | W3 | 4.4 | 6.8 | 7.2 | 0.67 | 11.9 | 13.5 | 11.1 | 0.54 | 4.3 | 6.2 | 1.2 | 1.12 |
| 19 | W4 | 4.4 | 6.6 | 7 | 0.62 | 11.9 | 13.5 | 11 | 0.56 | 4.2 | 6.1 | 1.2 | 1.10 |
| 20 | W5 | 4.4 | 6.9 | 7.1 | 0.67 | 11.8 | 13.8 | 10.6 | 0.72 | 4.1 | 5.9 | 1.2 | 1.05 |
| 21 | W6 | 4.4 | 7 | 7.1 | 0.68 | 11.8 | 14 | 10.4 | 0.78 | 4.2 | 6 | 1.8 | 0.94 |
| 22 | W7 | 4.4 | 7.1 | 7 | 0.68 | 11.8 | 13.6 | 10.6 | 0.67 | 4.2 | 5.9 | 1.3 | 1.03 |
| 23 | W8 | 4.4 | 7.2 | 7.1 | 0.71 | 11.9 | 13.7 | 10.7 | 0.67 | 4.4 | 6.3 | 1.1 | 1.17 |
| 24 | W9 | 4.4 | 7.2 | 7.1 | 0.71 | 11.8 | 13.4 | 11.1 | 0.52 | 4.3 | 6.2 | 1.2 | 1.12 |
| 25 | WM | 4.4 | 7.3 | 7.1 | 0.72 | 11.9 | 13.5 | 11.2 | 0.52 | 4.2 | 6 | 1.2 | 1.08 |
| Average | | 4.4 | 7.1 | 7 | 0.68 | 11.8 | 13.5 | 10.97 | 10.97 | 4.18 | 5.9 | 1.2 | 1.2 |

### 3.2. The Built Surface Effect on Thermal Surroundings and IR Results

The built surface temperature (ST) in relation to the effects of color and material upon the surroundings is presented in Figure S2 in part C of the Supplementary Materials. These results show that the textures and colors of built surfaces have an enormous role in the heat exchange, thermal emittance and absorption that impact ambient surroundings, which can significantly affect MRT. Therefore, surface properties and the impact of color significantly influence thermal comfort and heat balance outdoors, which can impact TSV and TCV substantially. The mean maximum and minimum temperatures of all measured artificial materials were remarkably close to the mean temperatures of water and tree parameters, thus these two parameters developing together may provide a stable-state environment. Concurrently, the thermal circumstances can be made more bearable by adjustment to the colors of hard materials nearby.

During the process of investigation, FLIR C3 thermal cameras [80] were used to detect heat radiation to identify the surface temperature of objects and people; infrared (IR) energy/heat was detected and converted into visual images at a resolution of 240 × 320 pixels. The resulting patterns and outcomes are shown in Figure 8.

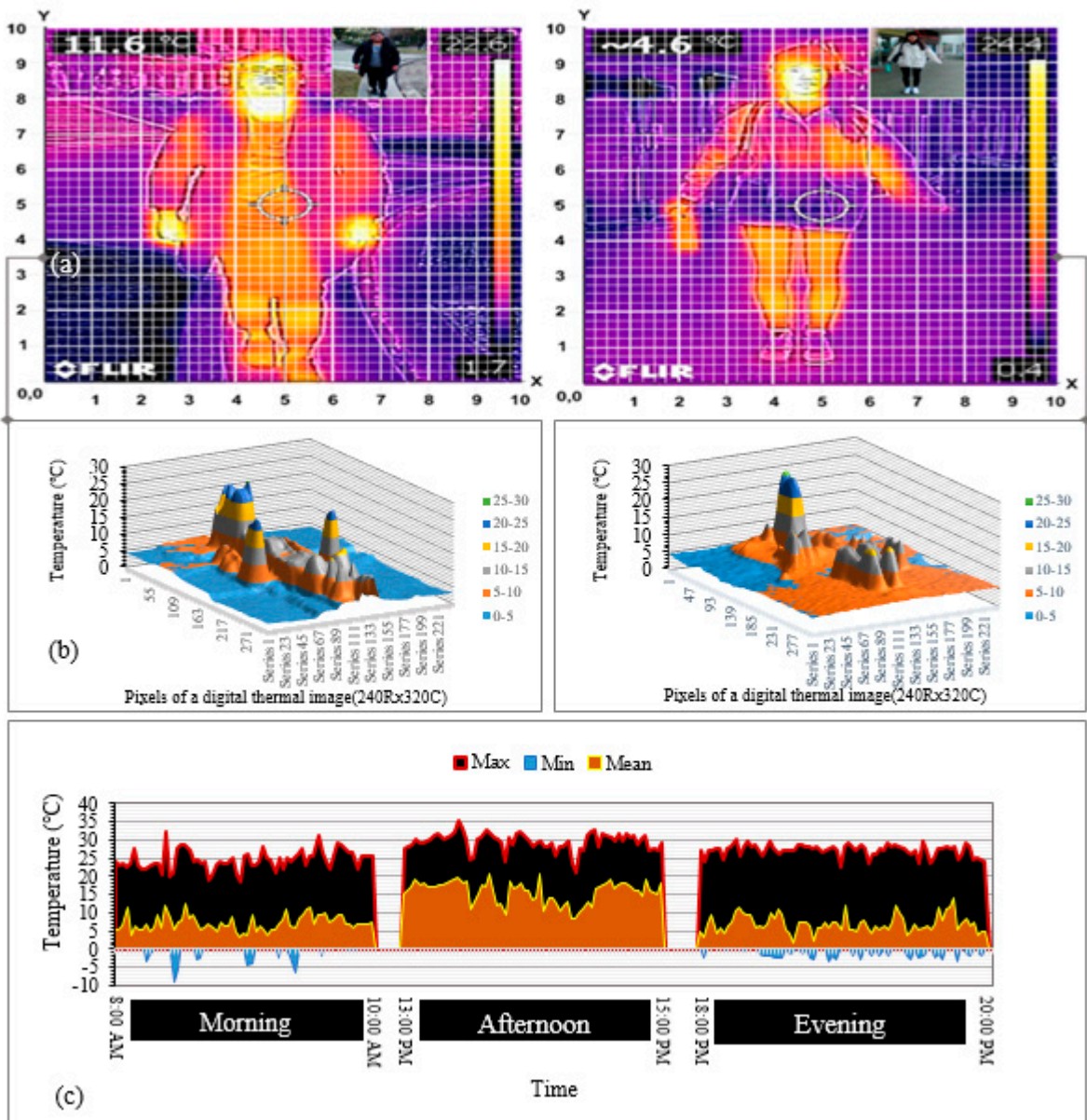

**Figure 8.** Heat flux distribution concerning the surface temperature of objects and people in three main zones. (**a**) A coordinated sample pattern of a visual image in pixels; (**b**) a sample of heat fluctuations analysis; and (**c**) the extracted results of analyzed cases in the morning, afternoon, and evening.

The effects of ambient temperature on heat distribution was determined in the morning, afternoon, and evening and findings compared with the influential parameters of the measured spaces, and questionnaire results. This facilitated better analysis of clothing insulation and thermal perception regarding body heat equilibrium and thermal comfort. It was found that the skin temperature of individuals indicated dynamic influence on OTC, especially when the maximum skin temperature in each sample signified an attempt to achieve body heat balance by increasing to normal skin temperature, which in the afternoon approached 33 °C or 91 °F. For all cases, the area of the eye closest to the nose was the measurably hottest spot on the face.

### 3.3. Thermal Sensation Vote (TSV), Thermal Comfort Vote (TCV), and Thermal Acceptability Vote (TAV)

### 3.3.1. Thermal Sensation Vote (TSV)

Figure 9a displays the frequency distribution of TSV of PCfa and PBWh respondents in winter. The outcomes revealed that the percentage of 'cold' sensation in winter for PBWh (45%) and PCfa (44%) was much higher than that of other sensations; the overall mean of the combined range between the main studied groups was found to be approximately 44.5%. Notably, the percentage value of 'very cold' sensation in winter for PBWh was, at 28%, two times larger than the corresponding figure for PCfa (14%), which indicates a significant difference in subjective sensation between groups. The linear regression model was established for the average thermal sensation vote (ATSV) and the PETI in winter. The respondents' thermal sensation votes were binned by the PETI interval of 0.5 °C in winter. The ATSV in each PETI bin operated as the dependent variable with every bin's main PETI center as independent variables. The relationship between ATSV and PETI is presented in Figure 9c.

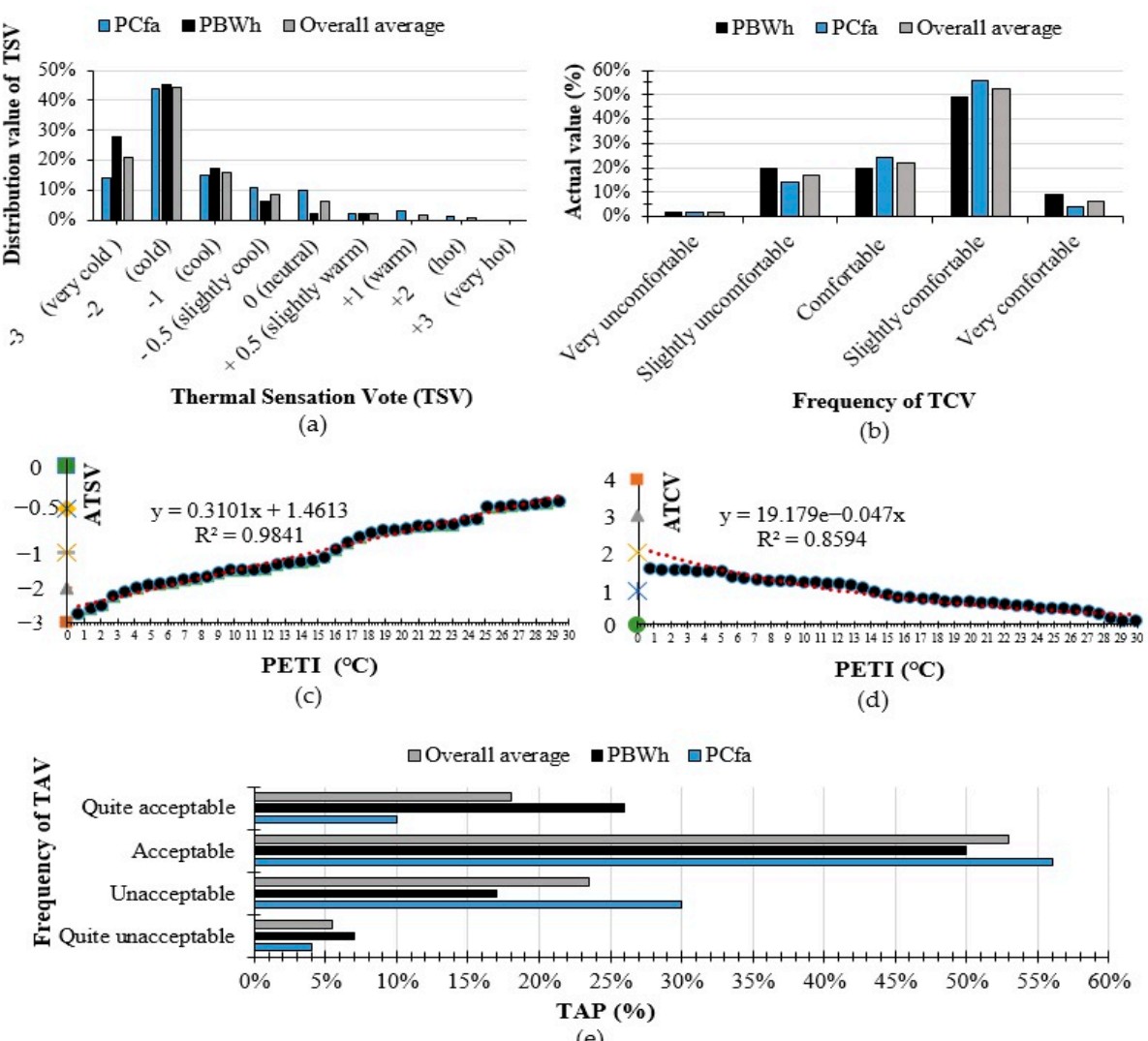

**Figure 9.** Thermal sensation vote (TSV), thermal comfort vote (TCV), and thermal acceptability vote (TAV). (**a**) Distribution value of TSV; (**b**) the frequency distribution of TCV; (**c**) the relationship between ATSV and PETI; (**d**) the relation between ATCV and PETI; (**e**) the frequency distribution of TAV at TAP.

### 3.3.2. Thermal Comfort Vote (TCV)

The frequency distribution of TCV is shown in Figure 9b. The frequency of 'slightly comfortable' votes was highest for both groups, PCfa (56%) and PBWh (49%), of all options in winter. The 'very uncomfortable' votes were the lowermost for both PCfa and PBWh, with both groups generating a result of 2%. The total estimation showed that 'slightly comfortable' and 'comfortable' votes were the highest proportion of the total votes in each group of PCfa and PBWh, at 80% and 69% respectively. This suggests that most people found the outdoor thermal surroundings of the activity spaces to be sufficiently comfortable. For PCfa and PBWh, the 'slightly uncomfortable' votes accounted for 14% and 20%, respectively, while the votes for 'very comfortable' were 4% and 9%, respectively. These are small proportions of the total combined vote. Simultaneously, the uncomfortable thermal sensation votes were higher for PBWh than for PCfa. The relationship between ATCV and PETI is shown in Figure 9d. It was observed that the smaller PETI value indicates a perception of greater comfort for all respondents.

### 3.3.3. Thermal Acceptability Vote (TAV)

Figure 9e shows the frequency distribution of TAV regarding thermal acceptability percentage (TAP). The results showed that both the highest (56%) and lowest (4%) thermal acceptance rates were for PCfa. However, the sum of 'quite acceptable' and 'acceptable' for PBWh was higher (76%), and the rate of 'quite unacceptable' and 'unacceptable' (24%) was lower than for PCfa at the same time, thus the acceptable thermal range for PBWh was greater in total. According to the statistical analysis, it was found that the ATAV (average of thermal acceptability votes) and PETI had a linear connection. This may be relevant for different thermal acceptability ranges in diverse temperatures.

### 3.4. Influential Elements

Statistical data analysis [49,52,53] was implemented for independent samples, in order to understand whether various measurement points had an interrelated effect on the respondents' thermal sensation votes. The T-test revealed that the investigation's chosen elements did have a substantial impact on people's sensation with regard to TSV. The mean heat stress of the R (road) together with W5 and W6 (built-up area) spots was higher than for HR (high-rise buildings—these can reduce temperature). The average heat temperature range for HR was between that of the spots above-mentioned (road and built-up areas) and that of the other green spaces (D2, D3, C2, B). Thus, they were lower than the road and the built-up areas, and higher than green spaces.

The highest range of heat balance alteration was observed in W6, while the lowest alteration range was for W3 with a single tree, lawn, and bushes. This shows that with increasing humidification, heat stress was reduced.

A strong dissimilarity effect on ATSV was found between the measuring points with high-density bushes in green spaces and single trees with shade-free spaces. Otherwise, there was no significant distinction between ATAV and ATCV in winter. Overall, the land surface characteristics had an essential impact on MRT and the surrounding temperature, in which reflection of the material may have positively affected the thermal environment. In winter, Ta is a crucial factor that affects PETI when openly exposed to solar radiation.

### 3.5. Gender Effect

Respondents' outcomes were analyzed for any differences between the genders that may be attributable to biological differences including metabolic process responses to diverse environmental circumstances.

The statistical T-test outcomes for both genders revealed that the ATSV values for females and males for PCfa and PBWh were 0.47, 0.5, 0.41, and 0.6, respectively. For both PCfa and PBWh males and females, $T_{exp.}$ was lower than $T_{critical}$ (sig. $\geq 0.05$), in which the difference of thermal sensation vote of the respondents was not significant. The same statistical method of T-test was implemented for the assessment of the respondents' thermal

comfort vote with regard to gender differences. The outcomes revealed that the ATCV value of females and males for PCfa and PBWh were 0.24, 0.52, 0.4, and 0.59, respectively. T-test analysis showed that sig. $\geq 0.05$ ($T_{exp} < T_{critical}$) for both males and females of PCfa and PBWh showed the same outcome for thermal sensation vote, thus gender difference had no strong impact on thermal comfort votes.

T-test analysis of the thermal acceptance votes by both genders revealed that the average values of thermal acceptance for females and males for PCfa and PBWh were 0.48, 0.52, 0.4, and 0.6, respectively, with a higher significant level of 0.05; $T_{exp}$. Was lower than $T_{critical}$, demonstrating that gender might not have any considerable effect on thermal acceptance vote. Nevertheless, it was found that PCfa females had a higher rate of acceptance of their thermal surroundings than PBWh females did, while for males, the result was the other way around in that PBWh male thermal acceptance was higher than that of PCfa males.

### 3.6. Thermal Insulation of Clothing, and Activity Level

Table 6 shows the average values of the thermal insulation of clothing, along with the respondents' average activity levels. The PBWh average activity (164 W/m$^2$) was slightly higher than that for PCfa (155 W/m$^2$); SD = 6.36, demonstrating that PBWh respondents undertook outdoor activities more than PCfa. Simultaneously, the thermal insulation value of clothing alterations for PCfa and PBWh was estimated at about 2.38 Clo and 1.72 Clo. The clothing layer analysis for the upper part of the body showed that 28% of PCfa males and 30% of females covered their bodies with three layers (SD = 1.54), which was higher than that of other upper clothing layer selections in winter, while this value for PBWh males and females was 30% and 19% respectively, with SD = 7.47. The analysis for the lower part of the body showed that 29% and 25% of PCfa males and females covered the lower part of the body with two layers, with SD = 2.91. The corresponding value was just 29% and 17% with SD = 10.6 for PBWh, indicating that the metabolic rate of PCfa participants demanded more layers of clothing than PBWh respondents. It is notable that the thermal insulation value of clothing decreased when the physiological equivalent temperature increased.

**Table 6.** The average value of thermal insulation of clothing and activity level.

| | Activity (M/m$^2$) | | | | | Clothing Insulation Value (Clo.) | | | |
| | PCfa | | PBWh | | | PCfa | | PBWh | |
| Items | Mean | SD | Mean | SD | Items | Mean | SD | Mean | SD |
| --- | --- | --- | --- | --- | --- | --- | --- | --- | --- |
| 1 | 62 | 11.313 | 23.5 | 4.949 | 1 | 0.125 | 0.018 | 0.216 | 0.041 |
| 2 | 51 | 2.121 | 15 | 5.656 | 2 | 0.311 | 0.034 | 0.374 | 0.039 |
| 3 | 46 | 4.949 | 5.5 | 0.707 | 3 | 1.021 | 0.278 | 0.517 | 0.051 |
| 4 | 9 | 3.535 | 3.5 | 0.707 | 4 | 0.338 | 0.011 | 0.172 | 0.006 |
| 5 | 12 | 4.242 | 4 | 1.414 | 5 | 0.039 | 0.005 | 0.246 | 0.001 |
| 6 | 9 | 3.535 | 2.5 | 0.707 | 6 | 0.191 | 0.017 | 0.147 | 0.003 |
| 7 | 15.5 | 9.192 | 7.5 | 6.363 | 7 | 0.351 | 0.03 | 0.047 | 0.007 |
| Average | 29.21 | 5.555 | 8.78 | 2.929 | Sum | 2.38 | 0.393 | 1.72 | 0.151 |

### 3.7. Behavioral, Physical, and Psychological Adaptation

Figure 10 visualizes the study findings for behavioral change of PCfa and PBWh respondents in percentages, based on climatic adaptation. The figure shows that 'putting on more clothes,' 'using an umbrella,' 'stay under the sun', and 'wearing gloves' were deemed grade A measures to enhance thermal comfort in winter. Findings suggest that improving environmental objects and states as far as practicable was more significant and much more comfortable for all groups than mental adjustments, in terms of achieving thermal comfort.

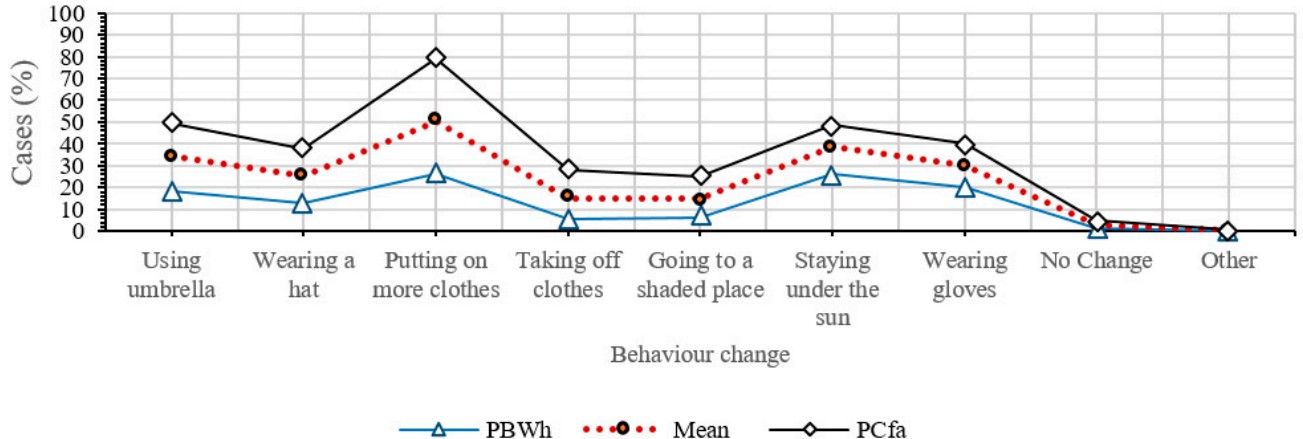

**Figure 10.** Behavioral change of PCfa and PBWh in percentage based on climatic adaptation.

To discover the likelihood of individuals taking various measures with a change in physiological equivalent temperature value, two independent and dependent variables were specified. These were the physiological equivalent temperature value and the measured behavioral category votes. Regression analysis [51] in SPSS, using the model of binary logistic regression [50], was carried out. Subsequently, the adapted model was examined via the test statistics of Hosmer–Lemeshow goodness-of-fit [53].

Table 7 shows the test outcomes with logistic regression equations. The confidence interval (CI for exp.) was set to 95%; if the likelihood *p*-value is smaller than the significant level ($\alpha = 0.05$), the dissimilarity among the predicted model value and the measured value is considerably high; if the likelihood *p*-value > $\alpha$, the dissimilarity among the predicted model value and the measured value is low, in which case the probable model has a more conforming definition.

**Table 7.** The models of logistic regression and estimated equations for behavioral conformity.

| Behavioral Conformity | Regression Logistic Equation (RLE) | Hosmer–Lemeshow Test | | | Exp (B) | 95% C.I Exp (B) | |
|---|---|---|---|---|---|---|---|
| | | Chi-Square | df | P | | Lower | Upper |
| Putting on more clothes | PCfa Logit(P) = 0.049PET + 0.131 | 6.375 | 6 | 0.383 | 1.050 | 1.002 | 1.099 |
| | PBWh Logit(P) = 0.004PET + 0.248 | 6.761 | 8 | 0.563 | 1.004 | 0.909 | 1.109 |
| Staying under the sun | PCfa Logit(P) = 0.005PET − 1.187 | 3.799 | 6 | 0.704 | 1.005 | 0.954 | 1.058 |
| | PBWh Logit(P)= −0.009PET − 1.510 | 6.566 | 8 | 0.584 | 0.991 | 0.870 | 1.130 |
| Using umbrella | PCfa Logit(P) = −0.008PET − 0.631 | 4.155 | 6 | 0.656 | 0.992 | 0.946 | 1.040 |
| | PBWh Logit(P)= −0.010PET − 0.369 | 12.975 | 8 | 0.113 | 0.990 | 0.895 | 1.095 |
| Wearing gloves | PCfa Logit(P) = 0.023PET − 1.722 | 9.042 | 6 | 0.171 | 1.024 | 0.965 | 1.085 |
| | PBWh Logit(P) =0.048PET − 0.589 | 9.927 | 8 | 0.270 | 1.049 | 0.949 | 1.159 |
| Wearing a hat | PCfa Logit(P) = 0.15PET − 1.111 | 2.667 | 6 | 0.849 | 1.015 | 0.965 | 1.067 |
| | PBWh Logit(P) = 0.097PET − 1.756 | 5.446 | 8 | 0.709 | 1.102 | 0.984 | 1.235 |
| Taking off clothes | PCfa Logit(P) = −0.052PET − 0.362 | 4.262 | 6 | 0.641 | 0.949 | 0.904 | 0.996 |
| | PBWh Logit(P)= 0.032PET − 2.233 | 10.316 | 8 | 0.244 | 1.033 | 0.888 | 1.201 |
| Going to a shaded area | PCfa Logit(P) = 0.020PET − 1.398 | 6.957 | 6 | 0.325 | 1.021 | 0.967 | 1.077 |
| | PBWh Logit(P)= −0.009PET − 1.759 | 4.249 | 8 | 0.834 | 0.991 | 0.859 | 1.143 |

The probable models revealed that all parameters of behavioral adaptation measures were significant. Additional consideration through Exp (B) analysis, particularly the Odds factor [81], revealed that with 0.5 °C reductions in physiological equivalent temperature value, the likelihood of 'wearing gloves' and 'wearing a hat' to enhance OTC rose more for PBWh than for PCfa participants, while 'staying under the sun', 'putting on more clothes',

and 'using an umbrella' was greater for PCfa than for PBWh. Behavioral frequency patterns under physiological equivalent temperatures thus demonstrate that most individuals preferred to attain thermal comfort via improvements to their physical environment, rather than attempting mental or psychological adaptation. In light of that, with decreasing physiological equivalent temperature value, people were more interested in using the open space environment for 'staying under the sun' in the wintertime, and for this, the rate was higher for PBWh, who had lighter clothing insulation values (1.72 Clo). This suggests that PCfa participants used higher clothing insulation values (2.38 Clo) and made more effort to obtain thermal improvements in this way, rather than adapt to the outdoor environment psychologically.

## 4. Discussion

### 4.1. TSV, Neutral Temperature (NT), and PET Correlation

Several thermal comfort studies have explored the correlation between TSV (thermal sensation vote) and NT (neutral temperature); NT was different in each study with diverse climatic zones [70,82]. The NT of the whole year in Wuhan, which is in the climate zone of Cfa (humid subtropical climate) is 24.0 °C ($\Delta T_{Winter}$ = 7.5 and $\Delta T_{Summer}$ = 6.7) [83]. It is 1.5 °C greater than that in Shanghai [84] and the present study's outcome concerning NPET (PCfa NPET; 19.7 °C and PBWh NPET; 22.6 °C), the NPET result of PBWh was close to that in the study from Canan et al. [30], who found a preferred PET of 30.5 °C and NPET of 21.9 °C in winter in the BSk climate of Konya, Turkey. This indicates that the NT value has a significant correlation with regional climate and its surrounding environments, based on people's behavioral adaptation physically and psychologically, which can be affected by the membership of different groups or even cultures.

According to the relevant study analysis, the NPET and the average activity of PBWh were greater than those of PCfa by about 2.9 °C and 9 W/m$^2$, respectively. At the same time, the clothing insulation of PCfa was higher (0.66 Clo). These findings may indicate that the adaptive behaviors and thermal perceptions of PCfa people in the Cfa climate of Wuhan were more flexible than those of PBWh people, most of whom came from a BWh (hot desert) climate. Meanwhile, the problem of psychological and physiological changes for PBWh females was more significant than for other cases. Accordingly, thermal sensation as a prompt to adjust the existing thermal environment was significantly interrelated with environmental parameter variations that strongly impacted people both physically and psychologically, with respondents consistently adapting their thermal sensation to adjust to the existing thermal surroundings.

### 4.2. Land Cover and Comfort Temperature

The outcomes of the present study revealed that for the investigated groups, participants did not perceive the neutral temperature as the most comfortable. The most comfortable, preferred PET in winter for PBWh was 30.1 °C, while for PCfa it was 27.6 °C—greater than NPET (approximately 7.5 °C). This is in line with the study conducted in Xi'an's cold region, which found a comfortable sensation at 29.7 °C [85].

Results suggest that the comfort/temperature of the microclimate measured in the current study was highly connected to the land cover, with the various colors of the built environment affecting the comfort of temperature modifications. It should be borne in mind that adjacent structures in various materials, with standard asphalt roads, also had an influence on the surroundings via the radiative process. Thus, landscape elements should be carefully considered in order to optimize the comfort of thermal circumstances, for example, the elements close to water surfaces can be affected by the water body's temperature. This can improve thermal comfort sensations in winter from cold to almost neutral temperature (NT). It can be confirmed that increasing water bodies can improve the heat balance and, consequently, thermal perceptions of the area. This effect can be verified by the work that has been done in Lingering Garden in Suzhou, China to improve the microclimate and thermal comfort adjustment in a CWHS zone [86]. Due to the effect

of land cover, the change in RH is in contrast with Ta, which is a factor crucial to comfort temperature. In this context, the DewPt is linked to RH, and based on the results of the lowest point (F2—pavement) and highest point (C2—mid-dense bushes); the less the DewPt increases, the smaller the amount of moisture in the air, which directly impacts how 'comfortable' it will feel outside, because a feeling of warmth is more clearly perceived than one of coolness. People tend to consider a warm sensation more comfortable than a cool feeling in winter, and vice versa in summer.

### 4.3. Mean Radiant Temperature and Acceptable Temperature Ranges (ATR)

According to the land surface analysis, mean radiant temperature (MRT) rose due to the hard paving and asphalt materials at R point (road); in contrast, grass and planted areas could reduce MRT and this can be distinguished from points B (low-rise grass), C1 (high-dense bushes), C2 (mid-dense bushes), D2 (high-rise tree with spread grass), and D3 (high-rise tree with soil), which is compatible with the study by Lobaccaro and Acero [87]. According to the surface temperature results for the built environment, the MRT of elements is affected by color difference, which can significantly influence Ta value; it can be considered a strategy to adjust environmental comfort temperature. Li et al. acknowledged that Ta affects the Clo value remarkably [88], showing that MRT in the direction of environmental Ta exchange has a significant impact on thermal sensation vote and subsequently affects the physiological equivalent temperature value at different points individually. This correlation influences ATR, which differs due to the physiological and psychological adaptation behavior, which relies on the thermal comfort perception of respondents. Based on the research observation, the ATR of PCfa was higher and more significant than that of PBWh. The range observed varied among groups, according to people's thermal perceptions; the ATR for PCfa ranged from 19.7 °C to 27.6 °C, while the ATR for PBWh ranged from 22.6 °C to 30.1 °C. The ATR result for PCfa resembled the findings of a study conducted in Tel Aviv [89] with a Csa climate [30], while the result for PBWh can be compared to findings from Taiwan [90], with hot summers and mild winters. This indicates that people of different regions have different thermal requirements, and that people who had long resided in the Cfa climate of Wuhan had greater acceptance of the low outdoor temperatures than PBWh.

### 4.4. Physical Characteristics and Gender

From the statistical analysis of the T-test, it can be inferred that trees with bare soil (D3) reduced the mean diurnal air temperature by 0.87 °C and grassed regions decreased it by 1.42 °C. These results are close to those from Lee et al. [91], who found the Ta reduction effect of trees and grasslands to be 0.6 °C and 1.1 °C, respectively, indicating the significant impact of selected physical environmental characteristics on people's perceptions concerning thermal sensation vote. In the present study, the surrounding concrete buildings with colors of yellow and grey had the greatest impact on thermal sensation, by about 6.46 °C and 5.02 °C, respectively, while the white colored concrete had a lesser effect (1.26 °C) on thermal sensation in a cold environment. The surface temperature of the physical environment influenced the radiant heat transfer experienced by pedestrians, in which the thermal sensation vote in unshaded points was greater than that of points with shadow, indicating the significance of SR affecting MRT and Ta in the cold season. This sensation occurred primarily because open spaces without shadow absorbed shortwave SR the most, which led to ambient Ta and GT being greater than those found in areas with shadowed spaces (as illustrated in Figure 7).

The investigation revealed that shadowed areas had the most impact on thermal sensation, more than on thermal comfort and thermal acceptance votes, in a cold period. The result was obtained from the studied thermal parameters of TSV, TCV, and TAV; however, the shadowed spaces are more tolerable and temperate in a hot season, which can change the TAV and TCV. The obtained outcomes align with the study by Lin [92], who declared that shaded spaces in the cold season and unshaded spaces in the hot season

affected human thermal discomfort. Accordingly, it has been found that SR correlated to PETI, which resembles Colter's findings [93].

The statistical T-test outcomes regarding ATSV and ATCV for different genders revealed that gender had no significant effect on thermal comfort and sensation votes. This outcome differed from the study carried out by Jin [94], who found that the neutral temperature of females was 3.1 °C higher than for males, signifying that the actual sensation vote (ASV) of females was considerably greater than that of males, which can influence ATSV. In all, the comparison of PCfa and PBWh females revealed that PCfa females had higher ATSV while the ATSV of the PBWh males was higher than for PCfa males; the effect of gender differences on thermal sensation can be linked to physiological and psychological variations and adjustment behavior. In this regard, the T-test for gender difference analysis revealed that the gender difference substantially impacted thermal acceptability votes. In general, females had higher thermal acceptability and more adaption behaviors to adjust thermal surroundings. Accordingly, the females' activity and clouding insulation values were lower than those for males; female/male values for PCfa respondents were 2.05% and 18.9%, respectively, while these values for PBWh respondents were 0.51% and 4.09%, respectively. The results suggest that females have a higher rate of adjustable behavior, both physiological and psychological, concerning the thermal acceptability of their environment.

## 4.5. Behavioral and Thermal Conformity

According to the clothing insulation results for PCfa (2.38 Clo) and PBWh (1.72 Clo), and taking into account the study by Wang [95], if the clothing value exceeds 1.25 Clo, people are more sensitive to cold environments. This indicates the importance of behavioral conformity to increase human thermal comfort with clothing adjustment, as verified by several investigations [96,97]. This illustrates the connection between the Clo and Ta; whenever the Ta value is reduced, the Clo value will increase—and vice versa. According to the study results and the significance of behavioral and thermal conformity both psychological and physiological, the correlation between microclimate, human behavioral conformity, thermal comfort, and their linkage to neutral temperature (NT) can be represented as in Figure 11. Differences in outdoor activity according to group and gender were also identified in this study. The level of outdoor activity for PBWh was higher than that for PCfa, and male activities were higher than females in total. The activity of male PBWh participants was higher than that of PCfa males. However, most individuals preferred to use the enclosed environment to adjust heat balance in cold conditions.

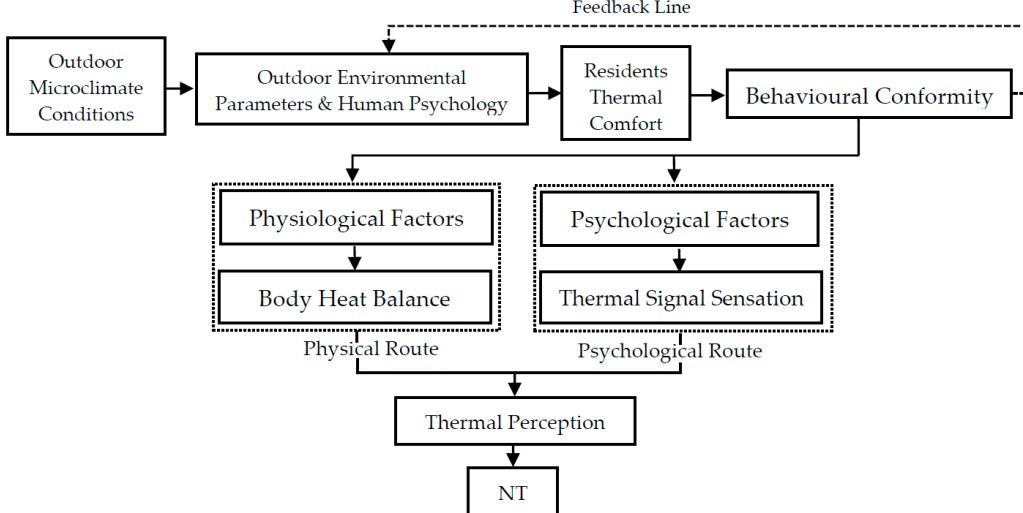

**Figure 11.** The correlation between microclimate, human behavioral conformity, thermal comfort, and their linkage to neutral temperature (NT).

## 5. Conclusions

This paper presents the results of a thermal comfort study conducted for the first time in the WHUT area of Wuhan between people with different biometeorological backgrounds in the KCC category of Cfa and BWh in winter. Individuals' psychological and physiological adaptive behaviors were considered. The major findings are as follows:

- In the outdoors, the behavioral conformity of occupants influences their NT under feedback routes physically and psychologically. It was found that in the Cfa climate of Wuhan, with the whole year NT equaling 24 °C, the NPET for PCfa and PBWh were 19.7 °C and 22.6 °C, respectively, in winter. This demonstrated that the NT value strongly correlates with regional climate and the surrounding environments, based on people's behavioral adaptation mentally and physically. Further analysis has also shown that the NT of thermal perception was not perceived by the investigated groups as most comfortable. The most comfortable PET in winter for PCfa was 27.6 °C, while for PBWh it was 30.1 °C, greater than NPET (approximately 7.5 °C). Furthermore, clothing adjustment has a notable impact on the NT as it is the most frequently-used method to control individual circumstances. The NT declined approximately 0.5 °C for winter when clothing insulation rose 0.1 Clo.

- The NPET and the average activity of PBWh were greater and more significant than those of PCfa by about 2.9 °C and 9 W/m2, respectively, while the clothing insulation of PCfa was higher (0.66 Clo). This indicates that subjective adaptive behaviors and thermal perceptions of PCfa people in the Cfa climate of Wuhan are more flexible than those of PBWh people, most of whom had a BWh climate background. Accordingly, it can be said that the behavioral conformity of PCfa people is more adaptive to the environment than that of PBWh, in winter. PCfa people are more tolerant of a cold environment than PBWh.

- The statistical ATSV and ATCV results within the T-test analysis revealed that gender, within the two specified groups, had no considerable impact on TAV and TCV under colder circumstances. Nevertheless, it was found that PCfa females had a higher level of acceptance of thermal surroundings than PBWh females, while the result was the other way around for males; PBWh males' thermal acceptance was higher than that of PCfa males. In all, there was a substantial relationship between TSV, NT, and PET. The thermal sensational prompt to adjust the existing thermal environment was significantly interrelated with variations in the environmental parameters. These variations affect people strongly, prompting physical and psychological responses, especially in terms of individuals seeking to adapt their thermal sensations to adjust to their surroundings. In this regard and among all cases, psychological and physiological changes by PBWh females as behavioral adaptation were more significant than in other cases.

- Overall, an individual's skin temperature reflected a dynamic influence on OTC. Additionally, statistical analysis of the behavioral, physical, and psychological adaptation among respondents revealed that 'putting on more clothes', 'using an umbrella', 'staying under the sun', and 'wearing gloves' were the preferred methods by which to enhance thermal comfort in winter. This signifies that improving environmental objectives as far as practicable is more significant and much more comfortable than mental adjustments to achieve thermal comfort.

- There is an explicit contrast between landscape variables and morphological/physical characteristics in urban surroundings. However, their combined effects are required for OTC improvement, which can substantially affect people's psychological and physiological behaviors. This study found a significant correlation between RH, DewPt, and Ta parameters of investigated landscape elements, and careful arrangement of these can greatly enhance the 'feels like' temperature and the sensation of human comfort. This aspect of environmental design can be controlled by planners and should be considered in order to achieve acceptable OTC. Notably, RH moves in the opposite direction to Ta, while DewPt has a greater tendency to move in the RH path. Dense

greening plays a substantial role. Furthermore, the land surface analysis revealed that outdoor thermal comfort and people's comfort sensations in the cold season can be improved by water body increment to almost NT.

These outcomes may be used in the design and planning for OTC to aid sustainable urban development in order to create thermally comfortable open-air surroundings in CWHS climate zones such as Wuhan.

In planned future work, the physiological differences between individuals that affect thermal responses will be modeled and added for thermal simulation. Accordingly, a thermal adaptive model will be added, based on KCC, along with psychological and physiological tests and analysis considering different groups in a CWHS region. Furthermore, the authors are incorporating urban surfaces and vegetation into microclimate simulation analysis. This is to support the current work by numerical methods to study the impacts that link urban built-up cover and vegetation and to improve urban thermal conditions, focusing their significant effect on people's physical and psychological responses.

**Supplementary Materials:** The following are available online at https://www.mdpi.com/2071-1050/13/2/678/s1, Figure S1: a strong relationship between the Ta and RH parameters of all measured points within three main time zones (morning, afternoon, and evening), Figure S2: the built surface effects on thermal sur-roundings, Tables S1, S2 and S3: thermal data regarding maximum, minimum, mean, and stand-ard deviation (SD).

**Author Contributions:** Conceptualization, M.M. and C.L.; Methodology, M.M.; Software, M.M. and C.L.; Validation, M.M., C.L., Q.D. and X.Z.; Formal analysis, M.M. and C.L.; Investigation, M.M., C.L., Q.D. and X.Z.; Resources, M.M.; Data curation, M.M. and Q.D.; Writing—original draft preparation, M.M.; Writing—review and editing, M.M. and C.L.; Visualization, M.M.; Supervision, C.L.; Project administration, M.M. and C.L.; Funding acquisition, C.L. All authors have read and agreed to the published version of the manuscript.

**Funding:** This research was funded by the National Natural Science Foundation of China (Grant No. 51608405); the Fundamental Research Funds for the Central Universities (Grant No. 203306001); and the China Scholarship Council (Grant No. 20160950013).

**Institutional Review Board Statement:** Not applicable.

**Informed Consent Statement:** Not applicable.

**Data Availability Statement:** Not applicable.

**Acknowledgments:** The authors thank the subjects who volunteered for this survey. This work was supported by the National Natural Science Foundation of China (Grant No. 51608405); the Fundamental Research Funds for the Central Universities (Grant No. 203306001); and the China Scholarship Council (Grant No. 20160950013).

**Conflicts of Interest:** The authors declare no conflict of interest.

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
