# Peer review of "A Field Investigation on Adaptive Thermal Comfort in an Urban Environment Considering Individuals’ Psychological and Physiological Behaviors in a Cold-Winter of Wuhan"

_sustainability, doi:10.3390/su13020678_

Round 1

Reviewer 1 Report

Dear Authors.

Yours paper is very interesting but you should change some things.

  1. In the text between words and number of bibligrpahy should  be space e.g. line 44, 49, 68 ... and rest.
  2. Why in the tables the first point is underline?
  3. In my opinion the figure 10 should be change, the line are too tick. When it will be thin and will be easier to read.
  4. Look at the line 416 and 418 - this needs to be corrected
  5. Lines between 474 to 480 hve different space than rest the text.
  6. The figure 15 should be change, this note applies "feebeek line"

Author Response

Dear Reviewer, 

I hope this email finds you very well.

Response to Reviewer1 comments file is attached and uploaded here in word format.

Please check and Thank you 

Reviewer 2 Report

Dear Authors,

Please address the following issues to improve the quality of the manuscript:

1- English writing is very poor and needs a significant modification. For example, in L 33-35, it is not a clear statement. In line 46 there is "*" sign and no other references to that. In L 55, there is ".." instead of ".". L81, "campus" should be written as "campuses" and etc...

2- Koppen Climate Classification (KCC) is mainly based on seasonal precipitation and temperature pattern. It was not intended to integrate psychological effect on the classification. I don't understand why there should be any study based on this particular classification vs psychological behavior of people....

3- Authors claim in L 97-99 "However, s far as we know, no previous research has focused on people's psychological and physiological behaviors according to the KCC system". If you google "psychological impact on thermal comfort", you will find tens of studies. For example, in heatwave studies, we know what group of people are vulnerable, what is the impact of heatwave on people behavior (check studies in UK regarding crime and temperature variation) and etc...

 4- This is at the best a case study and no one can use the results for other locations.

5- The logic behind the presenting many photos in the manuscript is not described. For example, what is the effect of those trees (D2 and D3) in Figure 4?

6- My main challenge is that the study focuses on a campus with majority of young and healthy people. What about other group of people?

7- What quality control standards have been used for questionnaires?

8- L204: What is the reason for showing this table and what is the ISO 7726? why these devices need to comply with that?

9- I couldn't find any significant relation between "Psychological and Physiological" state of participant and the climate.

Good luck

Author Response

Dear Reviewer, 

I hope this email finds you very well.

Response to Reviewer2 comments file is attached and uploaded here in word format.

Please check and Thank you very much

Reviewer 3 Report

In general terms, I consider that there is a lot of work done behind this article and I appreciate this work very much but, however it has been a bit difficult to read due to the multiple acronisms included as a results from different approaches developed and also due to the 25 measuring points which are indicated as lettersthrough the text in many cases.

The article is not well oriented as a proper reseach article.  In this sense the tittle, "Effects of Psychological and Physiological Behavious on Outdoors Thermal Comfort in Urban Environments in a Cold-Winter and Hot-Summer Climate" is not clear and it does not really indicate what it is being expressed through the article. 

Authors relate multiple measurements of meteorological variables done with different devices and multiple biometeorological indices expresing thermal comfort in different points. There is an important effort to assess how groups of people with different biometeorological backgrounds respond to different environments at a descriptive level.

In the Introduction section a good review of different thermal comfort indices is developed by authors and many of them have been computed and used in the paper. 

In Material and Methods, the study area is indicated correctly and methods use to collect data are reported (fixed-points observabtions, mobile observations, questionairesdigital infrarred thermometer and a hand heald infrared thermal camera)  

Data collection takes place three times per day in sunny days in winter  and continue for seven days. The criteria was to avoid climate conditions innapropiated what it is against the planned reseaech. It is not clear the period data were taken for. Moreover hot summer conditions have not been studied what indictaes again the tittle is not appropiated. The statistical methods were logistic regression, linear and multiple linear regression and T-Test accepted that thermal comfort-humans interactions respond to a liner model.

The aim is to link average thermal sensation, comfort and acceptability vote and thermal acceptability rate and physiological equivalent temperature with Neutral Temperature Zone NTZ, Comfort Temperature Range CTR, and Thermal Acceptability Vote TAV for two groups of people with different biometeorological backgrounds.  This aim should be define clearly at the end of teh Introduction section.

Results are described sequentially considering themal surrodings, landscape elements, built surfaces. However, in many cases the outcomes are obtained as a result of basic descriptions of data or without a proper discussion based on the outputs of the statistical models. In this sense, this article dos not respond to the aim of having a proper discussion section for each one of the results. 

I would recomend to authors to keep working on the hot summer frame to be able to compare also seasonal answers of two considered groups.

Author Response

Dear Reviewer, 

I hope this email finds you very well.

Response to Reviewer1 comments file is attached and uploaded here in word format.

Please check and Thank you very much 

Round 2

Reviewer 2 Report

Good Luck

Reviewer 3 Report

I consider the authors have gone through all my suggestions and the manuscript has been reduced and improved. 

This manuscript is a resubmission of an earlier submission. The following is a list of the peer review reports and author responses from that submission.

Round 1

Reviewer 1 Report

Dear Authors,

The manuscript prestented very interesting material. In the text are some misteaks, which should be changed:

1. Quotations should be separated by spaces from the text, e.g.: rope 68, 69, 99
2. The table description in line 243 should be above the table
3. The sentence "According to the connection of dew point (DewPt, ℃) with thermal comfort; its temperature can be calculated from RH as it has formulated below[56]:" should be given
4. In the figure in line 415 there is no figure c

Reviewer 2 Report

  1. Addition of a few illustrations especially where the applications are concerned to (in very specific way) capture the readers interest and imaginations, in the introduction section.
  2. Future trends and prospects of this article is missing and need to be elaborated further, stipulating the research directions.
  3. Every research results have recommendation for the following research in particular field so that following research can follow the earlier reports. Authr need to add the future prospective of his study. Author need to add the 2-3 sentences about future prospective based on his findings.

Reviewer 3 Report

Dear Authors,

Please consider addressing the following issues:

1- Check the abstract and rewrite that. It is not supposed to include materials from the introduction section.

2- Introduction should provide background and need for this study, which does not.

3-The study structure should be defined clearly AND should support the reasoning for that.

4- There are many data/photo in the manuscript which first of all are not necessary and also have very poor quality, for example photos in Table 9.

5-In many sections referencing is ignored and is very needed.

6- The manuscript must be checked for English grammar and writing.

Good Luck!